# Athlete Monitoring in Rugby Union: Is Heterogeneity in Data Capture Holding Us Back?

**DOI:** 10.3390/sports7050098

**Published:** 2019-04-27

**Authors:** Stephen W. West, Sean Williams, Simon P. T. Kemp, Matthew J. Cross, Keith A. Stokes

**Affiliations:** 1Department for Health, University of Bath, Bath BA2 7AY, UK; sw356@bath.ac.uk (S.W.); k.stokes@bath.ac.uk (K.A.S.); 2Rugby Football Union, Twickenham TW2 7BA, UK; simonkemp@rfu.com; 3Premier Rugby Limited, Twickenham TW1 3QS, UK; mcross@premiershiprugby.com

**Keywords:** rugby, athlete, monitoring, welfare, GPS, training, injury, performance

## Abstract

In an effort to combat growing demands on players, athlete monitoring has become a central component of professional sport. Despite the introduction of new technologies for athlete monitoring, little is understood about the practices employed in professional rugby clubs. A questionnaire was circulated amongst conditioning staff across the 12 Premiership rugby clubs to capture the methods used, relative importance, perceived effectiveness and barriers to the use of multiple different athlete monitoring measurements. Previous injury, Global Positioning System (GPS) metrics, collision counts and age were deemed the most important risk factors for managing future injury risk. A wide range of GPS metrics are collected across clubs with high-speed running (12/12 clubs), distance in speed zones (12/12 clubs) and total distance (11/12 clubs) the most commonly used. Of the metrics collected, high-speed running was deemed the most important for managing future injury risk (5/12 clubs); however, there was considerable variation between clubs as to the exact definition of high-speed running, with both absolute and relative measures utilised. While the use of such monitoring tools is undertaken to improve athlete welfare by minimising injury risk, this study demonstrates the significant heterogeneity of systems and methods used by clubs for GPS capture. This study therefore questions whether more needs to be done to align practices within the sport to improve athlete welfare.

## 1. Introduction

The monitoring of training load in sport is undertaken to maximise the potential for performance while minimising the risk of injury [1,2]. In recent years, there has been a proliferation in the use of technology in athlete management with practitioners across sport wanting to engage in a more scientific approach to monitoring their athletes [3]. While the use of increasingly complex technologies is growing, the use of more simple and cost-effective monitoring tools is also apparent, including the session Rating of Perceived Exertion (sRPE) method [4]. Session RPE is commonly used in rugby union as a monitoring tool for individual player internal load [5]. The potential utility of sRPE in the context of rugby union is evident given its ability to be used across multiple training modalities and its simplicity. Furthermore, it has also been shown that the tool is both reliable and valid when compared with other internal load measures such as heart rate and lactate [6,7]. The use of sRPE has been documented in rugby union as widespread, with 95% (*n* = 20) of the coaches of professional teams reporting its use [4]. Of this cohort, 95% considered sRPE an effective method for use in the management of individual player load with 63% reporting the measure’s effectiveness for injury prevention, 53% for illness prevention and 61% for enhancement of individual player performance.

Training load can be divided into two separate categories of load, namely, internal and external. External load is the physical work prescribed in a training plan, whereas internal load is the psychophysiological response of an athlete to that external load [8]. In the context of athlete monitoring, sRPE has been proposed and supported as one such measure of internal load and the response to training [4,9]. Despite the widespread use of sRPE within rugby union, there is a desire amongst sports scientists and coaches to harness the power of newly available player tracking technologies to enhance player welfare and performance. In recent years, the use of this technology has grown in the context of rugby union with extensive use amongst professional teams. However, although the use of Global Positioning System (GPS) technology is widespread, there is little understanding as to how each individual club collects, aggregates, reports and utilises GPS data, and to what extent methodological variation exists between clubs. To capture this information in the context of soccer, Akenhead and Nassis [1] undertook a questionnaire with sports science/medicine staff at professional soccer teams to understand more about their capture of training monitoring data. Of the 42 respondents, only 28 reported using sRPE, whereas all 42 reported the use of both GPS and heart rate to monitor their players. The main variables captured by clubs were accelerations, total distance, high-speed running, metabolic power and heart rate. The majority of the analysis of monitoring data was done using Microsoft Excel, while clubs reported lack of human resources and coach buy-in as the two greatest barriers to use within their clubs. Interestingly, only 20% of the respondents found GPS an effective measure of performance, with 23% considering the tool an effective measure for injury prevention. These findings are of particular interest given the widespread use of the tools as well as the time and money resources involved.

As the growth of GPS for athlete monitoring within rugby union continues, it is pertinent to understand how such data are collected, used and valued in this setting. Therefore, the aim of this study was to complete a survey of practitioners from each club in the English rugby Premiership to establish the principles of practice and the way in which GPS data are collected and to determine whether there is consensus amongst clubs.

## 2. Materials and Methods

Twelve staff members were asked to complete a survey online, one from each club in the English Premiership. The respondents included five sports scientists, three strength and conditioning coaches, one head of athletic performance and three heads of strength and conditioning, and had a mean years’ experience of 8 (±2) years. At the beginning of each questionnaire, a cover note was provided to the participants explaining the purpose of the survey and giving them an opportunity to ask the lead investigator questions regarding the study. Further verbal communication was undertaken with each of the questionnaire participants prior to distribution to ensure they were aware of the study requirements and purpose. Prior to the start of the survey, each participant was asked to provide consent for the study using a tick box at the end of the information page. Only once a participant had provided informed consent did the questionnaire populate with questions that were visible to the coaching staff. The study obtained ethical approval by the Research Ethics Approval Committee for Health (REACH) prior to the survey being distributed (Ref: 15/16 252).

The survey was reviewed by each member of the research team and changes regarding the content, structure and layout of the survey were made to ensure all necessary information was obtained. The questionnaire was then tested by one member of the sports science staff at another professional rugby club to assess its readability and content. Following feedback from this process, the questionnaire was sent to the 12 clubs participating in the English Premiership (the top domestic rugby union league in England). The final version of the survey comprised 25 questions (Appendix A). The first section contained questions regarding which monitoring tools were used by clubs to manage individual injury risk as well as the relative importance of each of those measures (Question 4). Section 2 included a series of questions about the use of GPS monitoring exclusively. That section required answers concerning the GPS system, version, unit type, software, measurement speed, metrics captured and relative importance of GPS for injury management and performance assessment. Finally, the remaining questions concerned the operational definitions used by clubs for variables such as high-speed running, as well as information regarding the barriers to using GPS within their individual team settings. The questionnaire took between 5 and 10 min to complete and was distributed to the respective staff members via email. If no response was received within one week, a reminder email was sent to the staff members. All 12 of the clubs had a staff member respond to the survey. The survey was designed and distributed using Bristol Online Surveys (now www.onlinesurveys.ac.uk). Data were collated and exported into a Microsoft Excel CSV file for analysis by the primary investigator. All data were presented as either median and interquartile ranges or frequencies of response, depending on the nature of the question asked.

## 3. Results

### 3.1. Tools for Injury Risk Management

To first establish the importance of a range of measures for injury risk management, each participant was asked to rank from 1 to 5 the importance of each measure outlined in Figure 1, with 5 representing a variable deemed “highly valued” and 1 representing “not at all valued”. All variables, with the exception of collision counts, player age and player experience, demonstrated a wide range of values with at least one club giving a value of 1 (not at all valued) and one giving a value of 5 (highly valued). Previous injury history was deemed the most valuable measure with a median response of 5, whereas GPS measures, collision counts and player age were the joint second most valuable tools, with a median of 4. All other measures carried a median of 3.

A second question was asked of the participants, which aimed to further understand the importance of GPS measures in the management of not only injury risk but also individual player performance. The results of this question are shown in Figure 2 and demonstrate the overall relative importance of GPS metrics for managing individual injury risk compared with the assessment in performance (median of 8 vs. 6). Furthermore, there was a wider spread of values associated with the use of GPS as a performance assessment tool (range: 1–10 vs. 3–10 for injury risk management) with one participant deeming it not at all useful.

### 3.2. GPS Collection Methods

To capture further information about the GPS measures collected and used by clubs, each participant then responded to a series of questions outlining information specific to their GPS use. Of the participants, 83% (10/12) reported using the CATAPULT system, whereas 16% (2/12) reported the use of STATSports. This can be broken down further to the OptimEye X4 (16%: 2/12), Sprint (16%: 2/12) and Openfield (50%: 6/12) versions for CATAPULT users and the APEX system for the STATSports users (16%: 2/12). Of the respondents, 25% used the X4 CATAPULT GPS units, whereas 58% used the S5 units. Both STATSports users had recently changed to the APEX units. A wide variety of computer software was used to analyse the GPS data with a large number of clubs using more than one software type. These included Openfield (33%: 4/12), Excel (42%: 5/12), Sportscode (8%, 1/12), Sprint (16%: 2/12) and APEX (16%: 2/12). The majority of clubs used units that were capable of 10 Hz recording speeds (92%: 9/12), whereas one team used 15 Hz units. The median number of units per team was 38 units with a range of 15 to 53. Only 42% (5/12) of clubs reported having enough GPS units to measure all players, whereas 33% (4/12) collected data for all senior players (non-academy) and 25% of respondents collected data for key players only. A question regarding barriers to data collection was also included. The most commonly reported barrier was the validity and reliability of GPS units (42%: 5/12), followed by lack of equipment (33%: 4/12), lack of staff to deal with the volume of data (16%: 2/12), lack of coach buy-in (16%: 2/12) and lack of consensus on best practice in GPS use (16%: 2/12). Further to these, 33% (4/12) of clubs reported no barriers to GPS data collection and one club reported “time to analyse data” as a barrier.

### 3.3. GPS Measures Utilised

A wide variety of measures were collected across teams (Figure 3) with the most commonly collected being distance in speed zones, and high-speed running distance in particular (100%: 12/12). Total distance was the next most commonly captured metric (92%: 11/12), followed by a count of sprints and metres per minute (83%: 10/12). Of the metrics collected, 42% (5/12) reported high-speed running as the most important metric for the assessment of performance, 25% (3/12) used metres per minute, 16% (2/12) reported total distance and 8% (1/12) reported the number of sprints. In addition, 25% (3/12) of respondents did not provide an answer to this question. In the management of injury risk, high-speed running was deemed the most important variable for 42% (5/12) of participants, followed by total distance (33%: 4/12) and accelerations/decelerations (8%: 1/12), with 33% (4/12) stating a combination of measures.

GPS measures can be collected in absolute terms (standard across all players, e.g., 5 m/s high-speed running) or relative terms (individual to each athlete, e.g., 70% of that player’s max velocity (Vmax)). In training, 33% (4/12) of respondents reported that measures were collected using absolute values, whereas 8% (1/12) of participants used relative measures only; 58% (7/12) recorded both relative and absolute. During matches, there was an even split between absolute and both measures being used with 50% (6/12) of participants using each. The measure high-speed running, which was reported as important for both the assessment of performance (42%: 5/12) and management of injury risk (42%: 5/12), was captured as an absolute value by 25% (3/12) of participants, relative by 58% (7/12) and both by 16% (2/12). For those reporting the use of an absolute high-speed running threshold, the values used were >5 m/s or >5.5 m/s, whereas in the relative group, values of 40–70% of Vmax, >49% of Vmax, >50% of Vmax, >60% of Vmax, >70% of Vmax and >80% of Vmax were used. In the classification of sprinting, absolute values of >6.7 m/s, >7 m/s and >7.5 m/s were used as well as relative values of >70%, >80% and >90% of Vmax. When asked if contact was captured during matches, 75% (9/12) reported that they did capture contact in games, whereas 25% (3/12) said they did not. Contact was measured using video analysis (66%: 8/12) and GPS (16%: 2/12).

## 4. Discussion

In the current monitoring practices of professional sports teams, there are a myriad of variables considered to be important for managing individual injury risk. While previous injury was determined to be the most highly valued measure for managing injury risk, GPS metrics were outlined as the most important monitoring tool for conditioning staff. Despite this, it was clear that the methods of data collection, the barriers to implementation of monitoring, the definition of key variables and the relative importance of metrics for performance assessment and injury risk management varied between clubs. This variation extended to all monitoring metrics with almost every measure being considered as “not at all valued” by one club and “highly valued” in another. This study presents an overview of the monitoring practices of professional rugby union clubs as well as definitions, methods, utility and perceived effectiveness of GPS metrics in the management of player welfare and tracking of player performance.

Injuries in sport have repeatedly been shown to produce negative consequences for team success [10,11,12] and, therefore, minimising the risk of injury is a key task for sports scientists, team medics and strength and conditioning coaches. In rugby union, a number of risk factors have previously been shown to be associated with injury risk, including the following: previous injury [13,14], player age [15], functional movement competency [16], and player load [14,17]. Despite this, little is known about how widely these measures are used in the individual management of injury and how they are valued amongst practitioners in an elite context. In this study, previous injury was deemed to be the most important risk factor with a median value of 5 (the highest possible score), which represented a “highly valued” measure (Figure 1). Although collision counts represent the next most highly valued measure (alongside GPS and player age: median score of 4), the capture of this metric is primarily dependent on time-consuming video analysis (66%: 8/12), while only 16% (2/12) of respondents used GPS-derived metrics to capture collision metrics. Interestingly, despite the reported widespread use of the sRPE monitoring tool [4], the measure represented one of the lowest scoring tools, with a median response of 3; however, values ranged from 5 (highly valued) to 1 (not at all valued). These values would appear to indicate the favourability of GPS-derived metrics for training load management over sRPE in the context of rugby union, despite the limited ability of GPS technology to fully represent the external load demands placed on athletes in collision sports due to the large amount of contact-based activity with little horizontal displacement [18]. Furthermore, the apparent favourability of external metrics comes in spite of recent work outlining the importance of internal loads in determining training outcome and thus the importance of these measures for athlete management [8]. With the advent of these newly available technologies, it appears that conditioning staff have been attracted to increasingly complex technologies to manage individual player injury risk, despite the concerns raised over the validity of the measures as well as the time-intensive nature of its use. The attractiveness of this technology and the fear of being “left behind” have potentially led to a belief that complexity (from a functional perspective, not a usability one) provides more favourable outcomes, which may not be the case if data quality is compromised for data quantity. Despite this, it is clear that GPS technology has established itself as an important tool for practitioners in professional rugby union, with coaching staff reporting a perceived greater effectiveness of GPS technology in the management of individual injury risk (median of 8) compared with the measurement of performance measurement (median of 6, see Figure 2).

With the advancement of GPS technology alongside the integration of tri-axial accelerometers into these units, the number of metrics available has become extensive, allowing practitioners to use only those that they deem most applicable and relevant for them. Of the numerous available metrics, distance in speed zones and high-speed running were reported as the most commonly collected, with high-speed running of particular interest. This metric can be recorded as a relative or absolute measure, with 58% (7/12) using relative measures, 25% (3/12) using absolute measures and the remaining 16% (2/12) using both. Of the relative measures, six different definitions of high-speed running were used, including 40–70%, >49%, >50%, >60% >70% and >80% of a player’s max velocity. Definitions of over 5 m/s and 5.5m/s were used for absolute measures. These findings demonstrate the substantial variation that exists between clubs when measuring what is considered the same GPS metric. In professional rugby union, running is the third most common mechanism of all injury, while running is also the most common mechanism of hamstring injury [19]. Given that hamstring injury is the most common training-related injury and given also that training is deemed a more “controllable” environment for injury prevention strategies compared to match play, this may offer a reason as to why high-speed running is considered so important in the management of individual injury risk. Current evidence suggests that a relationship between high-speed running load and injury exists [20,21,22], while it is also documented that well-developed physical qualities such as intermittent aerobic fitness may offset injury risk associated with rapid increases in high-speed running [22]. Knowing this, the monitoring of high-speed running within team sports has become widespread to minimise the risk of a potentially more controllable injury mechanism than that of contact. While the use of GPS data (in particular high-speed running) to measure performance is unlikely to be of huge benefit in rugby union, it may offer a surrogate measure to represent the ability of players to reach full speed, such as a line break or covering defensive tackle. In rugby league, faster speeds over 40 m have been associated with a higher number of tries scored [23], meaning the capture of such metrics may offer insight into player performance. It is likely however that key performance indicator metrics provided by statistic providers will offer a greater insight into player performance than GPS metrics alone. The next most commonly collected variables were total distance (92%: 11/12), count of sprints (83%: 10/12) and metres per minute (m/min) (83%: 10/12). These variables represent only a small proportion of the 17 that were provided in the questionnaire as well as several “Other” answers provided by the respondents. When asked to consider the importance of these variables for the purposes of performance assessment, high-speed running was deemed the most important by 42% (5/12) of respondents, whereas m/min was reported by 25% (3/12) of respondents. In the case of injury risk, high-speed running was deemed the most important variable (42% of respondents: 5/12), followed by total distance (33% of respondents: 4/12). This is consistent with previous work [1], and may be considered unsurprising given the more neuromuscular-orientated load associated with this type of exercise [24].

In the English Premiership competition, England’s highest level of club rugby, there are 12 clubs located across the country, each of which select, capture and utilise whichever monitoring methods they deem most appropriate to best manage their players. To gain insight into the systems, providers and methods used by clubs, the questionnaire requested the details surrounding the use of GPS so as to guide the project in targeting a group of clubs collecting and using similar data, to avoid comparison across differing systems. Of the 12 clubs, 83% (10/12) used the CATAPULT provider for their data collection. Although differences existed between unit types, each of the clubs using CATAPULT had monitors capable of recording at speeds of 10 Hz. On average, each club had 38 units at their disposal to capture GPS across their playing squad (mean and SD squad size: 57 ± 5), however this ranged from 15 to 53 units between different clubs, with four clubs stating a “lack of equipment” as a barrier to implementation. Interestingly, the most commonly cited barrier to further implementation or use of GPS data were issues concerning the validity and reliability of GPS data. These findings suggest that despite extensive work examining the validity and reliability of these measures as well as a number of systematic reviews [25,26], there is still concern amongst practitioners regarding GPS technology use in rugby union. One of the criticisms of GPS technology in the quantification of external loads in collision sports is the potential for the underestimation of demands due to the potentially large amount of impact activities with little horizontal displacement, including tackling, mauling and rucking [18]. Concerns raised over the validity and reliability of GPS data that were evident within the current questionnaire may be related to those criticisms over the capture of collision data. Despite these concerns, a 2016 study [27] has demonstrated good agreement between accelerometer-derived metrics Player Load, Player Load 2D and PLslow and manually coded collisions, concluding that practitioners can confidently use accelerometer-based metrics to quantify these aspects of play, in particular PLslow. While these findings are promising for the detection of collisions in games, differences between GPS systems and the youth population used in this study may limit the generalisability of the findings. In addition, the differentiation between different contact events would be useful, as opposed to the summation of all collision events. To ensure the validity and reliability of such measures in the context of one’s own club, it is encouraged that such validation studies be undertaken, to not only ensure the quality of data collected but also confirm or refute the findings of this study in an adult population. While such studies demonstrate the validity and reliability of contact-derived training load metrics, it is evident that this has not gained support within the elite club setting, with only two clubs using GPS-derived contact metrics and 42% (5/12) clubs outlining concerns over the validity and reliability of the units. Finally, when asked about the methods used in data collection by clubs, there is a lack of consensus with regard to the definition of specific GPS metrics, for instance high-speed running. Within this cohort, two different definitions were used when measuring the metric in an absolute manner and six different definitions when measured in a relative manner. This finding is supported in the work of Cummins et al. [28], who reported a lack of consistency in the reporting of speed zones and called for a consensus to be reached to aid comparison of demands within sports. While this study offers valuable insight into the methods used and perceived importance of monitoring metrics in professional rugby union, there are a number of limitations associated with the study. Despite capturing the opinions of one staff member across each of the clubs in the league, not only does this provide a relatively small sample size but also the views expressed may only represent that of the conditioning staff within the club, and the utility of monitoring metrics may be of different value to medical or other coaching staff. Another limitation concerns the possible answers that coaches could give within the survey. Although an “Other” option was given as an answer choice with a text box to elaborate, not all staff members availed of this option and, therefore, the use of qualitive interviews may offer more detailed insight in future studies. Finally, while these responses were representative of the staff at the time of answering, given the ever-evolving monitoring landscape, opinions and the relative importance of these metrics may have changed with the improvements in technology.

The current study documents the tools, methods and perceived value of monitoring variables collected by rugby union teams, in particular GPS technology. While this type of work has previously been undertaken in soccer [1] as well as in relation to sRPE specifically in rugby union [4], this is the first study to outline the relative importance of different monitoring variables in the context of performance measurement and injury risk management in rugby union. The current study also provides insight into the importance placed on GPS metrics by clubs, as well as the methods used to collect that data. What is clear from the questionnaire used in this study is that there is no consensus on best practice for GPS data collection in rugby union, with multiple definitions being used to collect the same variables. This study also highlights some of the difficulties associated with collecting this type of data, including lack of equipment, reliability and validity of the data as well as the staff required to deal with the volumes of data. Furthermore, this study outlines the wide variation in the relative importance placed by practitioners on certain metrics for injury risk management and performance measurement, with nearly every measure being highly valued in one setting but “not at all valued” in another. What is apparent from the current landscape in elite rugby union is that although the technological advancements associated with GPS use have made athlete monitoring more precise, the introduction of these increasingly complex methods of data collection may limit positive outcomes in athlete management until extensive work is undertaken to align and better understand how these metrics relate to individual injury risk and performance. Moving forward, for practitioners and researchers to maximise the value of these new technologies, consensus on the best available methods for data collection should be achieved. Attaining such a consensus would reduce the burden on conditioning staff for the collection, analysis and synthesis of such information, while also minimising the number of potential data collection points required from athletes daily. Moreover, given the range of resources available to elite rugby union practitioners to capture monitoring data, a minimum standard for athlete load monitoring should be suggested to ensure key elements of both external workload and internal response of athletes are captured.

## Figures and Tables

**Figure 1 sports-07-00098-f001:**
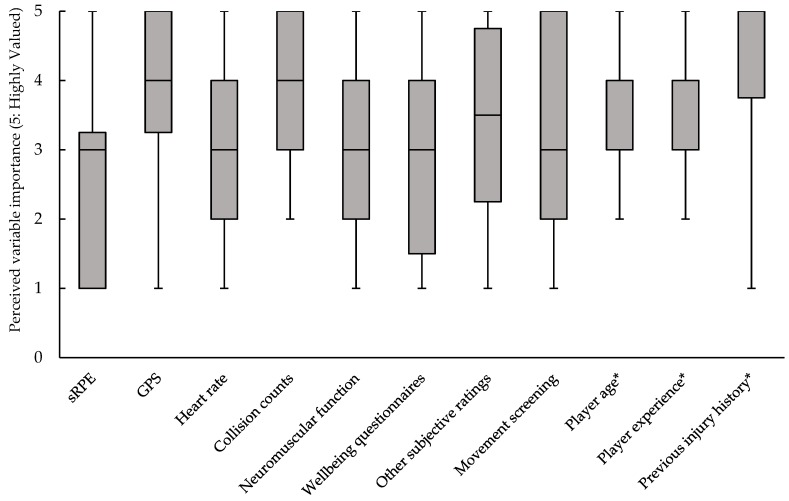
Box and whisker plot showing the median, interquartile range and range of values associated with responses to the question: “On a scale of 1–5, how highly do you value the following measures for the management of individual injury risk (where 5 represents highly valued and 1 represents not at all valued).” Grey boxes indicate the median and interquartile range, whereas the upper and lower end of the whiskers represent the lowest and highest observations, respectively. Variables exhibiting an asterisk presented the same median and upper quartile values, therefore the median is not visible. sRPE, session Rating of Perceived Exertion; GPS, Global Positioning System.

**Figure 2 sports-07-00098-f002:**
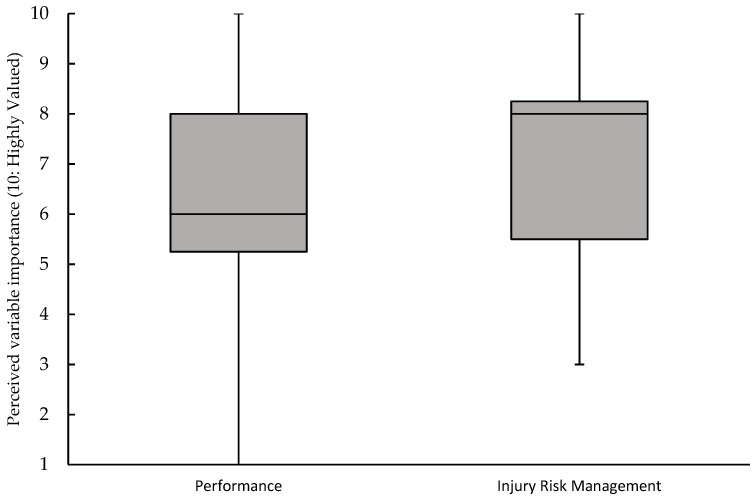
Box and whisker plot showing the median, interquartile range and range of values associated with responses to the question: “On a scale of 1–10, with 10 being the most important, how much do you value GPS data as a measure of player performance/individual injury risk management?” Grey boxes indicate the median and interquartile range, whereas the upper and lower end of the whiskers represent the lowest and highest observations, respectively.

**Figure 3 sports-07-00098-f003:**
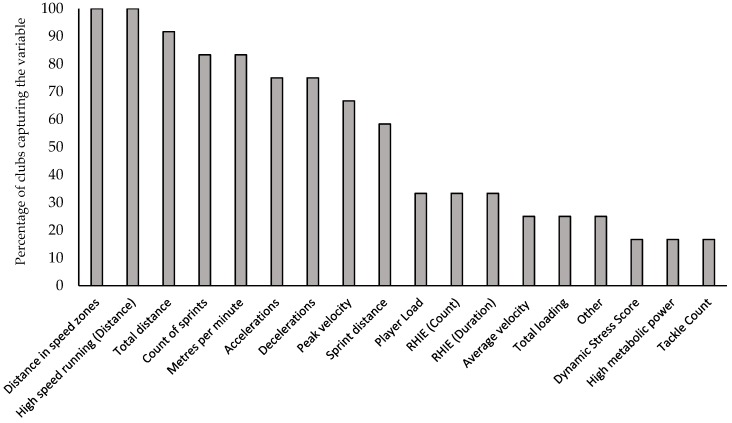
(x-axis) GPS metrics captured and (y-axis) percentage of teams recording these metrics. RHIE, Repeated high intensity efforts.

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
