# Peer review of "Athlete Monitoring in Rugby Union: Is Heterogeneity in Data Capture Holding Us Back?"

_sports, 2019, doi:10.3390/sports7050098_

Round 1
Reviewer 1 Report
This is a very well written manuscript of high importance and interest to the field. The paper clearly summarises the use of monitoring methods and perceived value of monitoring tools within the Rugby Union setting. Please see additional comments below.
Page 3, Results: Line 106 - it is stated that all variables with the exception of player age and player experience had values ranging from 1-5. Please confirm whether Collision counts should also be listed as an exception.
Page 5, Discussion: Lines 182-192 - quite repetitive of results, suggest removing from "Of those included..... high speed running in rugby union".
Page 6" Lines 233-243 - it would be good to see discussion around whether the literature in this field supports the perceived importance of the variables and their relation to performance or injury risk e.g. does high speed running correlate with actual rate of injury?
It would be good to see a clearer recommendation for moving forward. The current status of this field has been summarised and limitations identified, but what are your specific recommendations for progressing this field of research?
Author Response
Many thanks for your review, I have made the changes you suggested and I hope they address the points you suggested. Please see the attached new document with the changes I have made highlighted as comments.
Page 3, Results: Line 106 - it is stated that all variables with the exception of player age and player experience had values ranging from 1-5. Please confirm whether Collision counts should also be listed as an exception.
Thanks for spotting this, I have changed the text
Page 5, Discussion: Lines 182-192 - quite repetitive of results, suggest removing from "Of those included..... high speed running in rugby union".
I have removed the repitition
Page 6" Lines 233-243 - it would be good to see discussion around whether the literature in this field supports the perceived importance of the variables and their relation to performance or injury risk e.g. does high speed running correlate with actual rate of injury?
I have added a new section on this
It would be good to see a clearer recommendation for moving forward. The current status of this field has been summarised and limitations identified, but what are your specific recommendations for progressing this field of research?
I have added a new section on this
Reviewer 2 Report
This report presents valuable insight into athlete monitoring practices in elite-level sport.
I feel it is warranted to included at least some specific results (data) in the abstract.
I encourage the authors to include a section discussing the limitations of this project.
Please include titles for the vertical axis of figures 1, 2, 3.
Author Response
Many thanks for your review, I have made the changes you suggested and I hope they address the points you suggested. Please see the attached new document with the changes I have made highlighted as comments.
I feel it is warranted to included at least some specific results (data) in the abstract.
I have included some extra results
I encourage the authors to include a section discussing the limitations of this project.
I have added a limitations section
Please include titles for the vertical axis of figures 1, 2, 3.
I have added titles to the X-axis of each graph, thanks for spotting this.